# Comparison of Different Ex-Vivo Preservation Strategies on Cardiac Metabolism in an Animal Model of Donation after Circulatory Death

**DOI:** 10.3390/jcm12103569

**Published:** 2023-05-20

**Authors:** Stefano Mastrobuoni, Manuel Johanns, Martial Vergauwen, Gwen Beaurin, Mark Rider, Pierre Gianello, Alain Poncelet, Olivier Van Caenegem

**Affiliations:** 1Cardiovascular and Thoracic Surgery Department, Saint-Luc’s Hospital, Catholic University of Louvain, 1200 Brussels, Belgium; alain.poncelet@saintluc.uclouvain.be; 2Pole de Chirurgie Expérimentale et Transplantation, Institut de Recherche Expérimentale et Clinique (IREC), Catholic University of Louvain, 1200 Brussels, Belgium; martial.vergauwen@uclouvain.be (M.V.); gwen.beaurin@uclouvain.be (G.B.); pierre.gianello@uclouvain.be (P.G.); olivier.vancaenegem@saintluc.uclouvain.be (O.V.C.); 3Pole de Biochimie et Recherche Metabolique, Institue de Duve, Catholic University of Louvain, 1200 Brussels, Belgium; manuel.johanns@uclouvain.be (M.J.);; 4Cardiac Intensive Care Unit, Saint-Luc’s Hospital, Catholic University of Louvain, 1200 Brussels, Belgium

**Keywords:** heart transplant, donation after circulatory death, ex-vivo machine perfusion

## Abstract

Transplantation of heart following donation after circulatory death (DCD) was recently introduced into clinical practice. Ex vivo reperfusion following DCD and retrieval is deemed necessary in order to evaluate the recovery of cardiac viability after the period of warm ischemia. We tested the effect of four different temperatures (4 °C—18 °C—25 °C—35 °C) on cardiac metabolism during 3-h ex vivo reperfusion in a porcine model of DCD heart. We observed a steep fall in high-energy phosphate (ATP) concentrations in the myocardial tissue at the end of the warm ischemic time and only limited regeneration during reperfusion. Lactate concentration in the perfusate increased rapidly during the first hour of reperfusion and slowly decreased afterward. However, the temperature of the solution does not seem to have an effect on either ATP or lactate concentration. Furthermore, all cardiac allografts showed a significant weight increase due to cardiac edema, regardless of the temperature.

## 1. Introduction

Donation after circulatory death (DCD) was recently introduced into clinical practice for cardiac transplantation and may allow a significant increase (up to 48%) in the number of procedures performed per year [1]. Direct procurement (DP) is one of the two main established techniques for DCD heart retrieval. Compared to in situ normothermic regional perfusion (NRP), DP is simpler and more familiar to the transplant teams, does not require any intervention in the donor and has no ethical concerns compared to the reanimation of the heart in a person pronounced dead. However, following DP, ex vivo machine perfusion (EVMP) of the graft is deemed necessary in order to reassess the cardiac viability after a prolonged period of warm ischemia. EVMP further provides a key opportunity to improve cardiac recovery through target interventions. In this setting, the optimal solution and temperature during EVMP are still matter of investigation. The identification of the optimal protocol is, indeed, of paramount importance in order to fully exploit this donor pool, while reducing the risk for the recipient.

Currently, the only clinically available platform for cardiac EVMP is the Organ Care System (OCS, TransMedics; Andover, MA, USA), which allows a normothermic reperfusion. This system mainly relies on the inflow/outflow lactate levels, coronary flow and visual assessment of myocardial contraction of the unloaded heart which, in combination, act as markers of cardiac viability. Increasing lactate levels might reveal an ongoing anaerobic metabolism and are supposedly associated with poor cardiac recovery after transplantation [2]. However, lactate levels alone correlate poorly with primary graft dysfunction [3]. Moreover, normothermic perfusion may worsen ischemia–reperfusion injury (IRI) following a prolonged warm ischemic period, while a reduction in temperature may protect the heart via reducing the myocardial oxygen demand [4]. Furthermore, the OCS relies on 1.5–2 L of donor blood. However, the donor blood itself may have high concentrations of metabolites such as catecholamines following the prolonged ischemia, which may, in turn, adversely affect the cardiac recovery during EVMP.

Hypothermic oxygenated perfusion was previously implemented in DCD liver and kidney transplantation [5,6]. Hypothermic perfusion can maintain the heart in diastolic arrest (after cardioplegia infusion) and reduce the myocardial oxygen demand, as well as being simpler to implement compared to the system required to perfuse a normothermic beating heart with a blood-based solution. Thus, EVMP with cold crystalloid solution facilitates aerobic metabolism of the heart and low lactate levels, and provides functional and metabolic recovery superior to static cold storage [7]. Continuous hypothermic perfusion is also simpler to implement than EVMP with warm blood [7,8]. An animal study further suggested that mild hypothermia (between 25° and 30 °C) during EVMP improves functional recovery compared to profound hypothermia (20 °C) and normothermia in a rodent DCD model [4].

Despite the successful clinical implementation of normothermic EVMP, the optimal perfusion temperature is yet unknown. The aim of this study was, therefore, to compare the metabolic profile of the graft with four different temperatures during EVMP in an animal model of DCD heart.

## 2. Materials and Methods

### 2.1. Heart Procurement

This is a study on animals and we had the Ethical approval for this (reference: 2016/UCL/MD/024). The study was conducted in accordance with the guidelines outlined in the “Guide for the Care and Use of Laboratory Animals”, which were published by the National Institute of Health (NIH edition 85-23, revised 1985). We used 20 Landrace pigs weighing 50–60 kg. Anesthesia was induced via means of intramuscular injection of 6 mg/kg tiletamine and zolazepam (Zoletil; Virbac, Carros, France). Following tracheal intubation, inhalation anesthesia was maintained using a composition of enflurane (0–1.5%), nitrous oxide and oxygen. Electrocardiogram was continuously recorded. After median sternotomy, the heart was exposed, and the systemic blood pressure was monitored through a catheter in the distal ascending aorta. Heparin was administered intravenously at 600 IU/kg. After three minutes, ventilation was stopped and cardiocirculatory arrest defined when invasive systolic blood pressure was constantly (>2 min) below 50 mmHg. After a stand-off period of 20 min, the ascending aorta was cross-clamped and the heart flushed with 1 L of cold (4 °C) crystalloid cardioplegia (Custodiol©, Dr F. Köhler Chemie GmbH, Bensheim, Germany) delivered at a pressure of 65 mmHg via a 16-gauge cannula inserted into the proximal aorta. The heart was vented through an incision in the inferior vena cava and left atrial appendage and further cooled with cold topical saline solution. After termination of cardioplegia, the heart was retrieved in standard fashion and connected to the ex vivo machine perfusion. The timings of the different steps of the procedure are represented in Figure 1.

### 2.2. Preservation

The heart grafts were perfused on the HeartPort System© (Organ Recovery System, Itasca, IL, USA) (Figure 2).

This perfusion system was previously described in [9,10]. In summary, this system is a modified version of the LifePort© System used for kidney graft perfusion and preservation. It is a portable perfusion system where the explanted heart is lodged into a reservoir and suspended vertically via an injection cannula in the distal ascending aorta; the distal aorta is cross-clamped around the cannula. The cannula is connected to the perfusion circuit and allows antegrade coronary perfusion. The perfusate coming from the coronary sinus goes out of the heart through the open inferior vena cava, being recollected into the reservoir and recycled into the system. An external oxygenator and heat exchanger along the circuit allow for oxygenation and temperature control of the perfusate. The heart grafts were preserved for 3 h, with each graft randomly assigned to 1 of 4 groups (*n* = 5 in each group) with different perfusion temperatures and pressures (4 °C at 10–15 mmHg = Group A or deep hypothermia; 18 °C at 20–30 mmHg = Group B or moderate Hypothermia; 25 °C at 30–50 mmHg = Group C or mild hypothermia; 35 °C at 50–70 mmHg = Group D or Tepid Temperature) during EVMP. The pressure into the aortic root was checked at the beginning of the reperfusion in order to verify aortic valve competence and efficient coronary perfusion. The solution used was 1 L of KPS-1© (Organ Recovery System, Itasca, IL, USA) with a partial oxygen pressure between 150 and 200 mmHg maintained through means of the external oxygenator. The KPS-1© had the same composition as the University of Wisconsin (UW) solution originally formulated by Belzer [11]. The UW solution is a so-called intracellular hyperkalemic solution aimed to prevent cellular edema and is widely used for heart preservation. A constant temperature of 4 °C was maintained in Group A through means of the external cooler; in the other groups, the perfusate temperature was gradually increased to reach the target temperature over 30 min in order to ensure homogenous perfusion. In Group D (tepid temperature), KPS-1 was mixed with the donor whole blood in a 3:1 ratio. In this group, 300 mL of blood was slowly collected through a peripheral vein before the stop of the ventilation, at which point it was replaced with saline solution in order to maintain stable the circulating volume. In all cases, the temperature and coronary flow were continually monitored during EVMP.

### 2.3. Cardiac Metabolism

Cardiac metabolism was evaluated through dosage of high-energy phosphates in the myocardial tissue and serum lactate concentration. Five consecutive myocardial biopsies were conducted on the apex of left ventricle during the experiment. The baseline biopsy was taken in vivo on the donor pig before the onset of hypoxia. The 2nd biopsy was carried out after retrieval of the heart but before the onset of EVMP. The 3rd to 5th biopsies were carried out hourly after the beginning of EVMP. The timing of the biopsies is represented in Figure 1.

The samples were quickly frozen with liquid nitrogen and stored at −80 °C until further analysis. Myocardial tissue high-energy phosphate content was measured in neutralized perchloric acid extracts of the frozen myocardium, as previously described [9]. Adenosine monophosphate (AMP), adenosine diphosphate (ADP) and adenosine triphosphate (ATP) values were quantified after separation through means of high-performance liquid chromatography and based on the integration of unknown concentration peaks into known standard concentration peaks. All measurements were expressed in micromoles per gram, LV dry weight. Decreased intracellular ATP levels and higher AMP concentrations were shown to correlate with the severity of ischemia and poorer LV functional recovery [10].

Serum lactate concentration was also measured during the procedure. Baseline plasma lactate before the onset of hypoxia was measured in the systemic arterial blood and in venous blood from the coronary sinus. The 2nd lactate dosage was conducted at the onset of EVMP in both the perfusate solution on the ascending aorta (inflow) and the solution coming from the coronary sinus (outflow). The 3rd to 5th lactate levels were measured hourly after the beginning of EVMP in the same way. Lactate levels in the blood and perfusate solution samples were measured immediately with the iSTAT system (Abbott, Chicago, IL, USA). An increase in lactate concentration in the outflow perfusate revealed an anaerobic metabolism due to poor ATP regeneration following ischemia and was associated with poor recovery [2]. On the other hand, decreasing lactate concentration in the perfusate was a sign of myocardial viability [12].

### 2.4. Myocardial Edema

The cardiac grafts were weighted after retrieval and before the onset of EVMP (dry weigh) and at the end of the preservation period (wet weigh). The wet–dry weight ratio was estimated.

### 2.5. Statistical Analysis

Continuous variables were expressed as a mean and standard deviation. Differences between groups in the preoperative continuous variables were assessed with a two-way analysis of variance (ANOVA) with additional Bonferroni post-hoc tests for multiple comparisons. The evolution of high-energy phosphates and lactate concentrations over time and differences between groups were assessed with a mixed-effect longitudinal model for repeated measurements. Values of *p* < 0.05 were considered statistically significant. All analyses were performed using Stata/IC 15.1 (StataCorp, College Station, TX, USA).

## 3. Results

After withdrawal of mechanical ventilation, cardiac arrest (systolic blood pressure < 50 mmHg) supervened after a mean of 6.6 ± 3.4 min with no difference between groups. The time from the withdrawal of ventilation to the cardioplegia injection (warm ischemic time, WIT) was 26.6 ± 3.4 min, with no difference between groups (Table 1).

Allograft weight before EVMP was significantly larger in Group C; however, the ratio between allograft weight and body weight was similar between groups. Animals in all groups showed an increase in the allograft’s weight at the end of EVMP, and the wet–dry ratio was significantly higher for mild hypothermia (25 °C) and tepid temperature (35 °C) groups compared to the moderate hypothermia (18 °C) and deep hypothermia (4 °C) (*p* < 0.001) groups.

ATP concentration in myocardial tissue showed a steep fall at the end of the total ischemic period compared to the baseline values (*p* < 0.001), with no difference between groups (1.16 ± 0.48 nmol/mg) (Figure 3).

Furthermore, there was a significant decrease in ATP concentration during EVMP (*p* < 0.001), regardless of the perfusate temperature (*p* = 0.7). ADP concentration showed a similar trend, while the AMP concentration showed a slight increase towards the end of 3 h EVMP period (Figure 4).

Nevertheless, the AMP/ATP ratio did not change significantly over time, and no difference was recorded between the groups (0.29 ± 0.35 at the beginning of EVMP) (*p* = 0.2) (Figure 5).

Lactate concentration in the perfusate from the coronary sinus (outflow) increased significantly during the first hour of EVMP (*p* < 0.001) (4.5 ± 1.5 mmol/L); however, no significant difference was recorded between groups (Figure 6) (*p* = 0.1). Furthermore, we observed a trend toward a slow diminution over the second and third hour of EVMP. There was no correlation between lactate and ATP levels over time. Moreover, the ischemic time was predictive of neither lactate levels nor ATP levels during EVMP (*p* = ns).

## 4. Discussion

Limiting the ischemia–reperfusion injury is the key point in order to recover hearts following DCD. Reducing the cardiac metabolism, on the one hand, and allowing ATP regeneration, on the other hand, are the two main goals of EVMP. The ideal perfusate should, therefore, offer oxygen carrying capacity, metabolic substrates with oncotic properties and physiological pH.

This study sought to assess the impact of perfusion temperature during ex vivo machine perfusion on cardiac metabolism in a DCD animal model. In this study, we found that after a WIT below 30 min, ATP levels in myocardial tissue were remarkably low, and there was little regeneration of ATP, regardless of the temperature of perfusate during 3 h of EVMP.

Previous experimental studies found that the upper limit of WIT should be less than 30 min to ensure cardiac recovery [13]; nonetheless, a WIT of less than 30 min seems difficult to achieve in real clinical settings. The WIT is defined as the time between the withdrawal of life support and reperfusion of coronaries, and includes the (variable) time between the stopping of ventilation and cardiac arrest plus the stand-off period (that according to jurisdiction is between 2 and 5 min). In our protocol, we established that after cardiac arrest, there was a further 20 min stand-off period before cold cardioplegia injection (Functional warm ischemic time or FWIT). This choice was based on the previous clinical observation of a median time of 26 min between cardiac arrest and coronary reperfusion during DP [14]. Moreover, in this same experience, the median WIT was 37 min.

In this study, we also found that AMP, which was a precursor of ATP, only slightly increased over 3 h of reperfusion, particularly in the tepid temperature (35 °C) group. Furthermore, lactate concentration in the coronary sinus, which is a marker of anaerobic metabolism, kept increasing during the first 2 h of EVMP and started to slowly decrease afterwards, despite no correlation with ATP regeneration. It is probable that the first-hour increase in lactate in the outflow represents the washout of lactate produced in the myocardial tissue during the warm ischemic time, rather than a continued anaerobic metabolism during the EVMP.

Finally, we observed the development of myocardial edema as revealed through increased allograft weight after EVMP. Interestingly, we observed that the wet–dry allograft weight ratio, i.e., the ratio between the weight after and before EVMP, was significantly lower for Group A (deep hypothermia) than Group D (normothermia), with the other two groups’ values falling in between Group A and Group D. However, hypothermia groups failed to show improved metabolic profile, despite less edema, than the tepid temperature group.

The development of tissue edema following ischemia–reperfusion injury during EVMP is a matter of concern, as it may have an important impact on the cardiac function in the clinical setting. Several elements play a role on tissue infiltration: endothelial dysfunction, which appears early after ischemia (WIT) and cannot be controlled [15], as well as oncotic pressure of the solution and perfusion pressure during EVMP, which could both potentially be under control. Furthermore, hypothermia, through coronary vasoconstriction, may reduce the tissue infiltration compared to normothermia. Given that the oncotic pressure of the crystalloid solution is usually lower than the blood, perfusion pressure was kept low to reduce the hydrostatic forces and avoid vascular leakage. With increasing temperature, however, there is less vasoconstriction and higher metabolic demand; therefore, a higher perfusion pressure is needed. With tepid temperature and blood-based perfusion, physiologic perfusion pressure is required.

Reperfusion with autologous blood at tepid temperature has several potential advantages, such as highly effective oxygen transport, physiologic oncotic pressure and perfusion pressure. However, the autologous blood needed to prime the machine in the clinical setting (1.5 L) can only be retrieved after circulatory death; therefore, it carries high doses of lactate, cathecolamines, potassium and other metabolites reversed into the circulation from the whole body during the warm ischemic time, causing severe acidosis. Furthermore, platelets activation may cause microvascular obstruction and, nonetheless, at 35–36 °C the myocardial oxygen demand is higher, despite the unloaded state of the heart, than at lower temperatures. Normothermic reperfusion failed to show any significant benefit compared to static cold storage in the clinical setting of brain-death donation (DBD) [16]. Interestingly, in our study, the tepid temperature group showed similar weight increase and no remarkable benefit in terms of ATP regeneration. However, deep hypothermia, though it markedly reduces the metabolic demand, similarly lowers the activity of the membrane ion pumps that restore intracellular ionic homeostasis completely disrupted during WIT, which may ultimately alter the reparative process [17,18]. Therefore, considering that our group previously showed that hypothermic perfusion was nonetheless superior to cold static storage [9], the present study, as well as previous ones [4,19], seem to suggest that a mild-to-moderate hypothermia (18–25 °C) may sufficiently decrease myocardial oxygen demand, while allowing ATP regeneration coupled with the easiness of the EVMP set-up compared to the normothermia.

Furthermore, hypothermic crystalloid perfusion provides the opportunity to precisely control certain conditions, such as the composition of the solution. Metabolic substrates, such as amino acids and Krebs-cycle intermediates, may be easily added beforehand to the crystalloid perfusion in order to enhance ATP regeneration. Targeted medications, such as vasodilators, anti-inflammatory and anti-platelets agents, and post-conditioning agents, such as glyceril-trinitrate, erythropoietin and zoniporide, may be added to the cardioplegia solution during EVMP in order to reduce IRI and improve cardiac recovery [13,20].

Although hyperkalemic solutions, such as the UW solution and St. Thomas’ solution, are widely used for heart preservation in the setting of DBD, they may have a detrimental effect in the setting of DCD exacerbating IRI [21,22]. A normokalemic adenosine–lidocaine (A–L) cardioplegia may provide further protection from IRI. Lidocaine blocks the sodium fast channels maintaining diastolic cardiac arrest, while adenosine maintains a polarized membrane potential [21]. A–L proved superior to St. Thomas’ solution in terms of myocardial injury and endothelial function in experimental models of DCD [13,23]. A–L cardioplegia surely deserves further investigation in the DCD settings.

## 5. Conclusions

This study shows that after 26 min of warm ischemic time, ATP levels in myocardial tissue are very low. ATP regeneration is very limited during ex vivo machine perfusion. Lactate concentration keeps increasing during the first hour and descends slightly thereafter. All grafts showed weight increase, particularly with tepid temperature. Nonetheless, the temperature of perfusate seems to affect neither tissue energy recovery nor lactate production over time. The metabolic state of cardiac allografts following DCD is worrisome, and further strategies to improve cardiac recovery during ex vivo machine perfusion should be investigated.

## 6. Limitations

The main limitation of this study is the lack of a dynamic test on the allograft to show the extent of cardiac function recovery following EVMP. Therefore, we were not able to say if these allografts were ever able to generate a circulation. However, a Langendorff system and further heart transplantation in a swine model pose significant difficulties, not least the high number of animals needed. Nonetheless, with the machines currently available in the clinical setting, no functional tests can be performed, the heart is beating but unloaded, and evaluation of the allograft at the end of the ex- vivo perfusion is based on visual appreciation of contraction and the over-time trend in lactate production. The second major limitation is the small number of animals in each group, which means that the study is likely underpowered to detect significant differences between groups. Nonetheless, the discovery of very similar and worrisome metabolite concentrations in all groups regardless of the temperature are relevant.

## Figures and Tables

**Figure 1 jcm-12-03569-f001:**
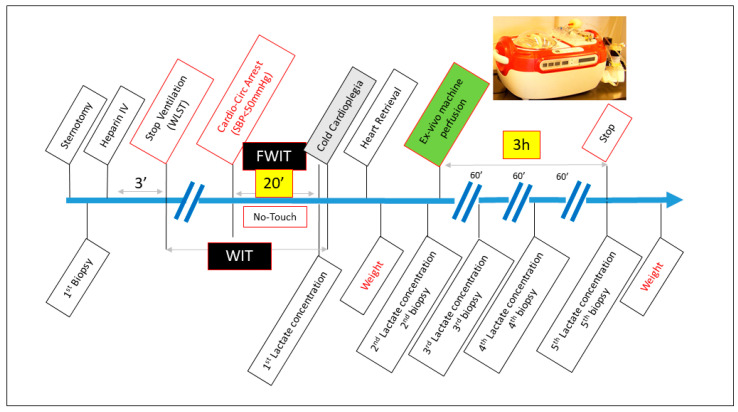
Schematic diagram of different steps and timing of experimental procedure. WLS: withdrawal of life support; WIT: Warm Ischemic Time; FWIT: Functional Warm Ischemic Time.

**Figure 2 jcm-12-03569-f002:**
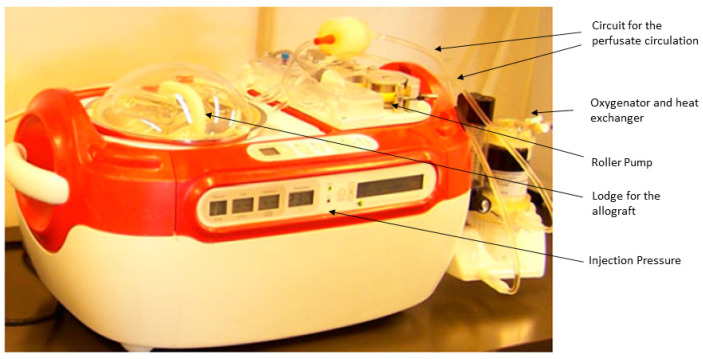
HeartPort© system (Organ Recovery Systems©, Itasca, IL, USA).

**Figure 3 jcm-12-03569-f003:**
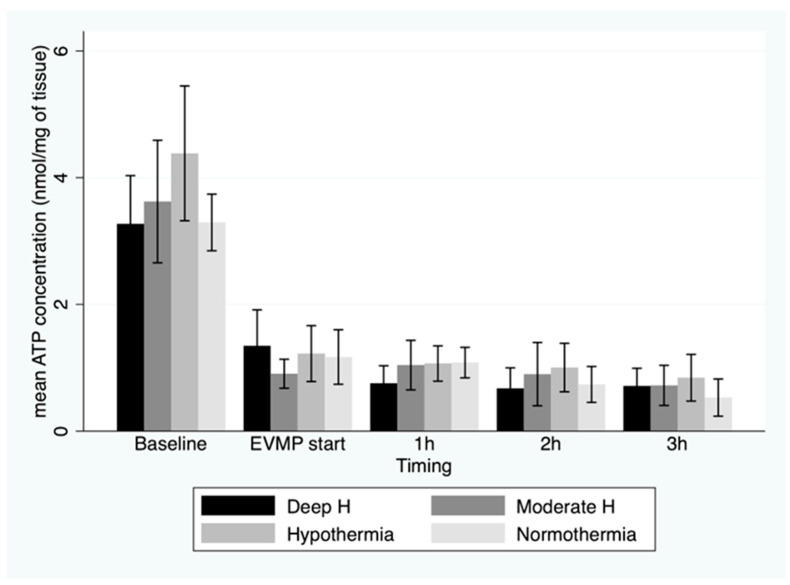
Mean ATP concentrations (with 95% CI) in myocardial tissue before onset of warm ischemia, at beginning of ex vivo machine perfusion and hourly during 3 h of perfusion by temperature group. EVMP: ex vivo machine perfusion; Deep H: deep hypothermia = 4 °C; Moderate H: moderate hypothermia = 18 °C; Hypothermia = 25 °C; Tepid Temperature = 35 °C.

**Figure 4 jcm-12-03569-f004:**
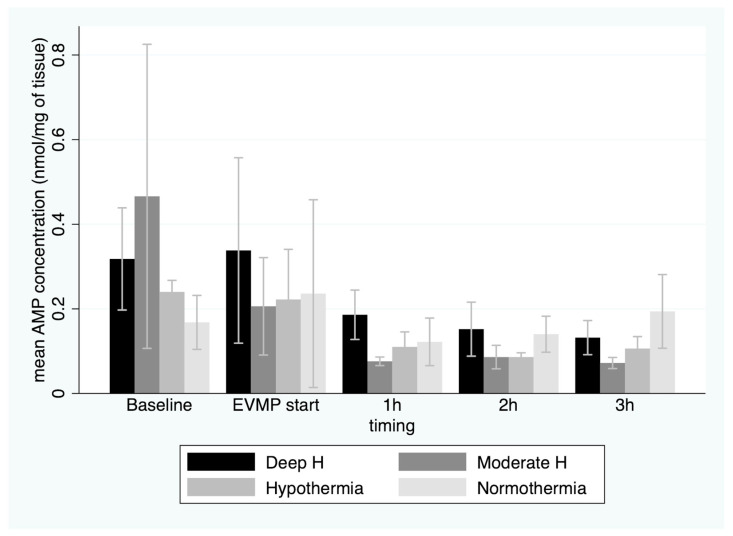
Mean AMP concentrations (with 95%CI) in myocardial tissue before onset of warm ischemia, at beginning of ex vivo machine perfusion and hourly during 3 h of perfusion by temperature group. EVMP: ex vivo machine perfusion; Deep H: deep hypothermia = 4 °C; Moderate H: moderate hypothermia = 18 °C; Hypothermia = 25 °C; Tepid Temperature = 35 °C.

**Figure 5 jcm-12-03569-f005:**
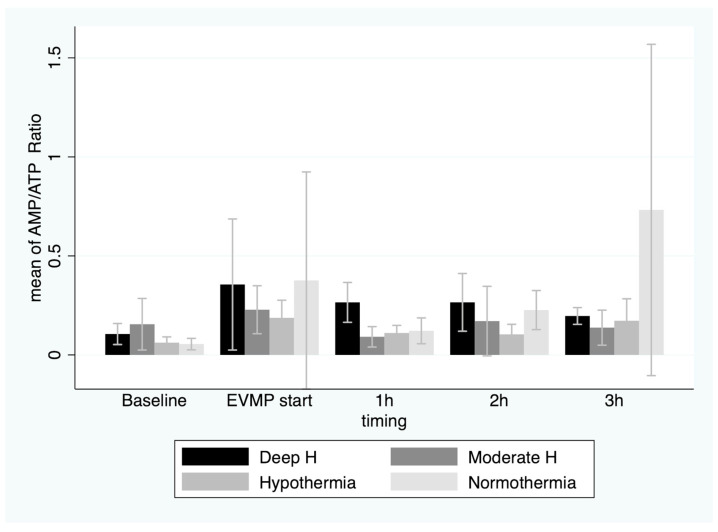
Mean AMP/ATP ratio (with 95%CI) in myocardial tissue before onset of warm ischemia, at beginning of ex vivo machine perfusion and hourly during 3 h of perfusion by temperature group. EVMP: ex-vivo machine perfusion; Deep H: deep hypothermia = 4 °C; Moderate H: moderate hypothermia = 18 °C; Hypothermia = 25 °C; Tepid Temperature = 35 °C.

**Figure 6 jcm-12-03569-f006:**
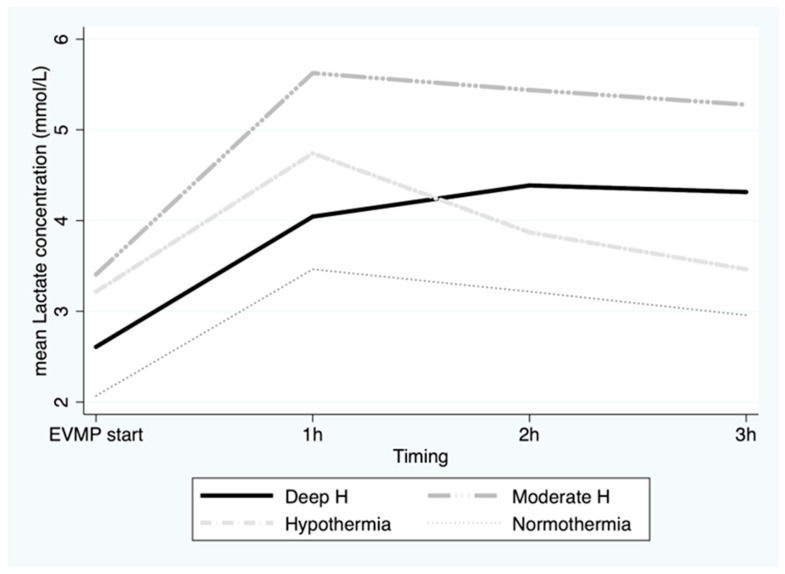
Mean lactate concentration in perfusate at beginning of ex vivo machine perfusion and hourly during 3 h of perfusion by temperature group. EVMP: ex vivo machine perfusion; Deep H: deep hypothermia = 4 °C; Moderate H: moderate hypothermia = 18 °C; Hypothermia = 25 °C; Tepid Temperature = 35 °C.

**Table 1 jcm-12-03569-t001:** Periprocedural Results by Groups.

	Group A (4 °C)	Group B (18 °C)	Group C (25 °C)	Group D (35 °C)	*p*
Time from WLS to cardiac arrest (min)	8 ± 4	7 ± 3	6 ± 3	7 ± 5	ns
Time from cardiac arrest to cardioplegia (FWIT) (min)	20	20	20	20	ns
Time from WLS to cardioplegia (WIT) (min)	27.8 ± 4.2	26.6 ± 2.5	26.2 ± 2.6	26.0 ± 4.6	ns
Time from cardiac arrest to EVMP (min)	39 ± 4 *	41 ± 4	46 ± 5 *	39 ± 4	0.06 *
Allograft weight before EVMP (grams)	279 ± 46	281 ± 16	364 ± 26	284 ± 53	0.01
Allograft weight after EVMP (grams)	299 ± 43 *	394 ± 42	428 ± 45 *	383 ± 100	0.02 *
Allograft weight wet–dry ratio	1.1 ± 0.0 *	1.4 ± 0.2 *	1.2 ± 0.1	1.3 ± 0.1	0.01 *

WLS: withdrawal of life support; FWIT: Functional Warm Ischemic Time; WIT: Warm Ischemic Time; EVMP: ex vivo machine perfusion; * the lowest *p*-value for pairwise comparison.

## Data Availability

The data presented in this study are available on request from the corresponding author. The data are not publicly available due to further ongoing research.

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
