# Peer review of "Comparison of Different Ex-Vivo Preservation Strategies on Cardiac Metabolism in an Animal Model of Donation after Circulatory Death"

_jcm, 2023, doi:10.3390/jcm12103569_

Round 1

Reviewer 1 Report

1. The introduction might use some work; greater context about the issue at hand would be helpful.

2. The results of the hemodynamic variables be incorporated into the study.

3.  Provide an explanation and data on vascular resistance.

Author Response

  1. The introduction might use some work; greater context about the issue at hand would be helpful.

We accept this reviewer’s suggestion (as well as Reviewer 4) and we have modified the Introduction.

  1. The results of the hemodynamic variables be incorporated into the study.

Before cardiac arrest, all animals were hemodynamically stable and we monitored only the invasive arterial pressure. During ex-vivo machine perfusion we could only check the injection pressure into the aortic root. Unfortunately we do not have any other data to show. No change has been made to the text.

  1. Provide an explanation and data on vascular resistance.

We did not address the issue of vascular resistance. Nonetheless, we explain in more detail the choice of different perfusion pressures along with the temperature. We have modified the Discussion.

Reviewer 2 Report

The manuscript is well written and the topic of graft preservation is important for further recipients' condition. Furthermore, the development of graft transfer strategies in animal models enables further safe application in humans. In your study interesting is the lack of correlation between ATP, AMP, lactates  and the perfusate temperature. The possible significant difference resulting in further outcome may be a consequence of edema.

According to your results could you explain which temperature is the best for the graft recovery? Do you think that the machine perfusion seems to be much better than the cold storage and short ischemic time? Could you discuss further possible directions in perfusate fluid composition?

Author Response

The manuscript is well written and the topic of graft preservation is important for further recipients' condition. Furthermore, the development of graft transfer strategies in animal models enables further safe application in humans. In your study interesting is the lack of correlation between ATP, AMP, lactates  and the perfusate temperature. The possible significant difference resulting in further outcome may be a consequence of edema.

According to your results could you explain which temperature is the best for the graft recovery? Do you think that the machine perfusion seems to be much better than the cold storage and short ischemic time? Could you discuss further possible directions in perfusate fluid composition?

1- We are glad for the interest that this reviewer shows towards our work. Our data unfortunately do not allow for a definitive response on the best temperature during Ex-vivo machine perfusion. Nonetheless, considering the easiness of moderate cold perfusion compared to the normothermia (with autologous blood), and our previous research that showed better results of hypothermic EVMP compared to static cold storage, we are keen to say that mild to moderate hypothermia (18-25°C) is probably the best temperature and further research should compare EVMP with this temperature versus normothermia and static cold storage. Further, the composition of the perfusate is a very important point. Different authors question whether hyperkalemic cardioplegia is counterproductive for cardiac recovery. Specific solutions used during EVMP over few hours may play indeed an important role in recovery. We have discussed these issues in length in the Discussion that has been so modified.

Reviewer 3 Report

The authors examined the impact of different temperatures of ex-vivo reperfusion on the cardiac metabolism in a porcine model of DCD. They found that perfusion temperatures affected neither ATP nor lactate, but the weight of allografts increased in all temperature groups, particularly with normothermia.

The study is overall well designed and presented, although the findings are not affirmative regarding to which temperature is preferable it does expand our knowledge pool in this field.

I was wondering if the authors could rationalize the different pressures they use during the perfusion period.

Author Response

The authors examined the impact of different temperatures of ex-vivo reperfusion on the cardiac metabolism in a porcine model of DCD. They found that perfusion temperatures affected neither ATP nor lactate, but the weight of allografts increased in all temperature groups, particularly with normothermia.

The study is overall well designed and presented, although the findings are not affirmative regarding to which temperature is preferable it does expand our knowledge pool in this field.

I was wondering if the authors could rationalize the different pressures they use during the perfusion period.

Response:

We are glad that our paper pleased this reviewer. We agree that unfortunately we were not able to give a definitive answer to the question of the optimal temperature. Nonetheless, we argue, based on this paper and others, that probably the best temperature lays in the 18-25°C range. We also explain the rationale behind the different perfusion pressure going with the different temperatures.  We have modified the Methods and Discussion.

Reviewer 4 Report

There is no conflict of interest in reviewing this manuscript submitted by Stefano Mastrobuoni, et al. entitled as “Comparison of different ex-vivo preservation strategies on cardiac metabolism in an animal model of donation after circulatory death.”

Summary

Clinical application of ex vivo or ex situ machine perfusion (EVMP) of cardiac allografts is gaining attention worldwide in the recent decade. Clinical data show comparable outcomes, mostly short term, using ex vivo warm perfusion to those using traditional cold storage for donors from donation after circulatory death (DCD), donation after brain death (DBD), of extended donation criteria, or of extended ischemic time. The outcomes refer to survival of the cardiac allograft or the recipients after heart transplantation. However, some studies challenges the benefit of  EVMP in terms of graft function at the temperature setting of 35-36 °C.

The present porcine, Landrace pigs, study compared the metabolic outcomes of DCD porcine cardiac allografts under 4 different temperature settings, 4°C, 18°C, 25°C and 35°C, respectively, using EVMP for 3 hours. The evaluating outcomes include left ventricle tissue quantities of adenosine monophosphate (AMP), diphosphate (ADP), and triphosphate (ATP) using high performance liquid chromatography. Comparative measurement outcomes include weight of cardiac allografts, serum lactate at the arterial end and in the coronary sinus around the EVMP circuit at specific time points.  No statistically significant differences of ATP levels between those tissues from different perfusion temperatures.

The study might provide additional information in the understanding of the physiologic impact on the cardiac allografts using this EVMP. Other points that need to be addressed are as follows:

Major--

Materials and Methods:

1.      Page 2, line 84: for group D the temperature of 35°C might be described as “tepid temperature” rather than "normothermia".

2.      What was the biopsy location of left ventricle in the experiments, apex? Lateral wall? or others? Considering the risk of in-homogeneous myocardial perfusion, multiple biopsy at different parts of LV might truly reflect the metabolic outcomes.

3.      Page 3, lines 106-109: for quantification of AMP, ADP and ATP, the author used only HPLC method, Other quantification methods might also be adopted as a reconfirm the accuracy of HPLC.

Results:

1.      For a better understanding of the timing described in the study and in the Table 1. The authors should compose, as a supplementary file, a diagram to show each period of timing during the DCD process: WIT, FWIT, etc.

2.      for the contemporary increase of lactate in the perfusate at 1 hr after perfusion in all allografts, could this be a washout period due to the WIT insult?

Discussion:

1.      In Discussion, page 7, lines 181-206: the first 3 paragraphs might be removed to Introduction.

2.      Please add a paragraph describing the limitations of this study.

Minor:

1.      Page 3, lines 125-126: for (dry weigh) and (wet weigh)--> should that be (dry weight) and (wet weight)?

2.      Figure legend for figure 1: please explain each abbreviation at the end of the legend, e.g. EVMP, H and ATP, etc.

minor changes are recommended as mentioned in the comments.

Author Response

Responses: 

  1. Page 2, line 84: for group D the temperature of 35°C might be described as “tepid temperature” rather than "normothermia".

We accept this suggestion and we have changed with “Tepid Temperature” all over the manuscript.

  1. What was the biopsy location of left ventricle in the experiments, apex? Lateral wall? or others? Considering the risk of in-homogeneous myocardial perfusion, multiple biopsy at different parts of LV might truly reflect the metabolic outcomes.

All myocardial biopsies were taken at the LV apex as the easiest accessible part of the LV throughout the experiment. We have specified this in the Methods.

  1. Page 3, lines 106-109: for quantification of AMP, ADP and ATP, the author used only HPLC method, Other quantification methods might also be adopted as a reconfirm the accuracy of HPLC.

We used this technique already in the past (see references 3 and 6) with consistent measurements therefore we did not apply other methods unfortunately. The text has not been changed.

Results:

  1. For a better understanding of the timing described in the study and in the Table 1. The authors should compose, as a supplementary file, a diagram to show each period of timing during the DCD process: WIT, FWIT, etc.

We thank this reviewer fo this suggestion and we have added a diagram (Figure 1) with the different steps and timing of the procedure.

  1. for the contemporary increase of lactate in the perfusate at 1 hr after perfusion in all allografts, could this be a washout period due to the WIT insult?

This is a very good point and most likely this is the case. During the WIT probably all ATP is consumed and the anaerobic metabolism starts with production of Lactate and release during the first period of EVMP. The lactate is washed-out subsequently but unfortunately, we have not seen a regeneration of ATP in the following hours. We have modified the text in the Discussion.

Discussion:

  1. In Discussion, page 7, lines 181-206: the first 3 paragraphs might be removed to Introduction.

We accept this suggestion and we have moved those paragraphs to the Introduction that has been changed accordingly.

  1. Please add a paragraph describing the limitations of this study.

 We thank the reviewer for this suggestion and we have added a paragraph reporting the 2 main limitations of the study (the absence of a functional test at the end of the EVMP and the small number of animals in each group).

Minor:

  1. Page 3, lines 125-126: for (dry weigh) and (wet weigh)--> should that be (dry weight) and (wet weight)?

We have corrected the text.

  1. Figure legend for figure 1: please explain each abbreviation at the end of the legend, e.g. EVMP, H and ATP, etc.

We have explained the abbreviations and added a legend to all figures.

Round 2

Reviewer 4 Report

No further comments to the authors.